# Modulation of Thermal Insulation and Mechanical Property of Silica Aerogel Thermal Insulation Coatings

Zhigang Di [1,2], Shengjun Ma [1,2], Huanhuan Wang [1,2], Zichao Guan [1,2,3,*], Bingjie Lian [1,2], Yunpeng Qiu [1,2] and Yiming Jiang [3]

1   CNOOC Energy Technology & Servicers Limited Key Laboratory of Corrosion Protection for Offshore Oil Industry, Changzhou 213016, China
2   CNOOC Changzhou Paint & Coating Industry Research Institute Co., Ltd., Changzhou 213016, China
3   Department of Materials Science, Fudan University, Shanghai 200082, China
*   Correspondence: zghydxhxgzc@163.com; Tel.: +86-0519-83270027

**Abstract:** In this paper, high-performance silica aerogel (SiO$_2$ aerogel) thermal insulation coatings were obtained and profited from the excellent thermal insulation capability of SiO$_2$ aerogel. The comprehensive properties and thermal insulation mechanism of the coatings were investigated via Scanning Electron Microscope (SEM), Fourier Transform Infrared Spectroscopy (FT-IR), contact angle, and temperature difference tests. Results showed that there was a contradiction between thermal insulation and mechanical property in this coating after the addition amount and proportion of silica aerogel, hollow glass microsphere, glass fibers, aqueous acrylic emulsion, and dispersing agents were optimized carefully. When the mass ratio of hollow glass to SiO$_2$ aerogel microspheres was 1:1, the overall performance of the coating was the best with thermal conductivity of 0.050 W/(m·K) and adhesive strength of 1024 kPa.

**Keywords:** silica aerogel; thermal insulation coatings; thermal conductivity; adhesive strength; formulation design





## 1. Introduction

As we all know, the oil and natural gas industries are energy intensive, and it is necessary to deal with heat fluid in different operation stages during oil exploitation, transportation, processing, and storage. In order to improve production efficiency and reduce costs, it is very important to minimize energy loss. Thermal insulation is an important solution for energy conservation and emission reduction in the petrochemical industry [1–4]. The existing thermal insulation materials (stone wool, glass wool) and thermal insulation structures often have defects such as complex construction procedures, corrosion under the insulation layer (CUI) [5,6], and heat loss caused by an incomplete protection layer. In recent years, thermal insulation coating has been widely studied and applied because of its advantages such as stable thermal conductivity and workable performance, which means it can be coated on any confined space and special-shaped work pieces. For offshore oil production platforms, pipelines or equipment usually work in a harsh corrosive environment with high humidity, high salt, and a high risk of typhoon damage [7–9]. In addition to the thermal insulation function, a high anti-corrosion function is required. Therefore, it is necessary to develop coatings with both thermal insulation and anti-corrosion functions. More specifically, the coating is supposed to withstand continuous cyclic thermal stress to provide thermal insulation function and maintain good mechanical performance to provide anti-corrosion function. As such, in addition to functional performance, a key criterion is adhesion to base metal surfaces. In summary, the design of insulating coatings that are thermal resistant, corrosion inhibited, and applicable to pipelines in service remains an area of active research [10–13].

$SiO_2$ aerogel is considered to be the best thermal insulation material at present because of its unique spatial network structure, high porosity, and low thermal conductivity. Much recent effort has focused on embedding aerogel within polymeric media to achieve a combination of low density and high porosity, which impedes heat transfer and reduces thermal conductivity to below 0.06 W/(m·K) [14–21]. The relationships between the thermal conductivity and volume fractions, densities, sizes, and interfaces of $SiO_2$ aerogel microspheres was the key research point. Li et al. [19] prepared thermal insulation coatings using $SiO_2$ aerogel and hollow glass beads as main function fillers, a mixed solution of silica sol and pure acrylic emulsion as film former, and this thermal insulation coating had a low thermal conductivity of 0.045 W/m·K. In Wang's work [20], they used sodium silicate as a film former, $SiO_2$ aerogel, and hollow glass beads as thermal insulation fillers to prepare thermal insulation coating and studied the influence of infrared shading agents on the coating performance. The results showed that the addition of rutile $TiO_2$ and titanium sol could significantly increase the extinction coefficient of the coating, and made the thermal conductivity decrease to 0.032 W/m·K. Wei et al. [21] Introduced carbon nanofibers with a mass fraction of 20% into $SiO_2$ aerogel to prepare carbon nanofiber-reinforced $SiO_2$ aerogel composites with enhanced high-temperature resistance. However, there are common defects such as low adhesive strength and slight cracking. In extreme cases, the coating peels off due to external damage, such as from typhoons or personnel movement. The thermal insulation performance is contradictory to the mechanical performance. Little attention has been paid to the comprehensive performance of thermal insulation coating.

With the above background, in order to develop a thermal insulation coating that can attach to the surface well, and possesses low thermal conductivity, a thermal insulation composite coating was prepared using $SiO_2$ aerogel microspheres, fillers, aqueous acrylic emulsion, and certain kinds of agents. The modulation between the thermal conductivity and mechanical strength is modified by adjusting the coating formula. The thermal insulation mechanism of the composite coating is also discussed.

## 2. Materials and Methods

The materials used in this work are listed in Table 1.

**Table 1.** Material list.

| Material | Type | Manufacturer |
|---|---|---|
| Hydrophobic $SiO_2$ aerogel | AG-S 15 | Shanxi Yangzhong new material Co., Ltd., Taiyuan, China |
| Hollow glass bead | VS 5500 | 3M Co., Ltd., Saint Paul, MN, USA |
| Nano zirconia powder | EFUZR-D30D | Zhengzhou Cheng Ao chemical products Co., Ltd., Zhengzhou, China |
| Fiberglass | FG-11 | Yancheng Ailiwei fiber products Co., Ltd; diameter 11 μm, Yancheng, China |
| Ceramic fiber | 1430 | Zhejiang Jiuwei refractory material Co., Ltd.; granularity < 1 mm, Hangzhou, China |
| Wetting agent | APM 95 | Dow Chemical, Midland, MI, USA |
| Defoamer agent | BYK-025 | BYK-Chemie GmbH, Wesel, Germany |
| Dispersant | CTD-7101A | CNOOC Changzhou Paint & Coating Industry Research Institute Co., Ltd., Changzhou, China |
| Waterborne acrylic resin | CTD-6803 | CNOOC Changzhou Paint & Coating Industry Research Institute Co., Ltd., Changzhou, China |
| Film-forming agent | Texanol-12 | Eastman, Shanghai, China |
| Thickening thixotropic agent | F01-V | Lehmann & Voss & Co., Hamburg, Germany |
| Deionized water | - | made by the laboratory, 16.68 MΩ·cm |
| Carbon steel board | Q235 | bought on the market |

### 2.1. Preparation of SiO<sub>2</sub> Aerogel Slurry and Thermal Insulation Coatings

2.1.1. Preparation of $SiO_2$ Aerogel Slurry

The moderate hydrophobic $SiO_2$ aerogel powders were firstly mixed with about 4 mL KH550 wetting dispersing agents, silane coupling agents, and deionized water in a 250 mL beaker at room temperature, which was then dispersed at high speed (600–1000 r/min) with

agitator for 12 h to obtain a uniform aerogel slurry. During the preparation process, KH550 was hydrolyzed to produce 3-aminopropyl trihydroxysilane with Si–OH and Si–NH$_2$ groups, which could graft to the surface of SiO$_2$ aerogel to form hydrophilic SiO$_2$ aerogel.

### 2.1.2. Preparation of Thermal Insulation Coatings

Fibers, powder fillers, and thickening thixotropic agents were added to the slurry and stirred. After the fibers were dispersed evenly, waterborne acrylic resin and film-forming agent were added. Finally, hollow glass beads and defoamer agents were added and stirred at low speed (300–500 r/min) to obtain the silica aerogel thermal insulation coatings. The mass ratio of each component was listed in Table S1.

### 2.2. Performance Tests

**Composition and morphology characterization:** The functional groups on the surface of SiO$_2$ aerogel power during the modification of SiO$_2$ aerogel were detected by Fourier transform infrared spectroscopy (FT-IR, Cary600, Agilent, Santa Clara, CA, USA). The contact angles of SiO$_2$ aerogel power before or after modification were measured by a contact angle tester (OCA25, Dataphysics Instruments GmbH, Filderstadt, Germany). The microstructures and dispersion status of the microspheres were observed using a scanning electron microscope (SEM, SU3500, Hitachi Company, Tokyo, Japan). Fiber dispersion in the coatings was estimated by a metallographic microscope (DM2700M, Leica Company, Wetzlar, Germany).

**Thermal conductivity tests:** The thermal conductivity of the coating was measured by a transient plane source method using the thermal conductivity measurement instrument (TC3200, XIATECH Company, Xi'an, China). The coating was evenly coated into the specific mold, dried in an oven at 50 °C for 24 h, and then demolded. The size of the sample was 50 mm × 50 mm × 3 mm. The thermal environment temperature was defined at 25 °C.

**Temperature difference tests:** The prepared coating was directly sprayed on the hotplate surface (with a coating area of 150 mm × 75 mm and thickness of 4 mm). The temperature of the hotplate was set at 120 °C, and then the temperature of the coating surface was measured utilizing a thermal couple sensor (TES-1310, TES Company, Taiwan, China). Finally, the temperature difference between the coating surface and hotplate surface was calculated. The schematic diagram was shown in Figure S1.

**Mechanical performance measurement:** Prior to the measurement, the prepared coating was sprayed on a carbon steel plate, and the coating thickness was 2 mm. The adhesion strength between the coating and carbon steel was determined with the pull-off test for adhesion in accordance with the standard GB/T 5210-2006 (EN ISO 4624: 2002), using an AGS-500NT electronic universal testing machine (SHIMADZU, Kyoto, Japan). The surface of the coated spruce sample and the steel-roller dolly with a diameter of 20 mm were connected by ALTECO 110 Super Glue.

## 3. Results and Discussion

### 3.1. The SiO$_2$ Aerogel Slurry

To avoid the decrease in thermal insulation performance caused by the infiltration of the polymer into the internal pores of SiO$_2$ aerogel power, the hydrophobic SiO$_2$ aerogel power was selected as the raw material in this work. Figure 1a shows an SEM image of hydrophobic SiO$_2$ aerogel in the waterborne coating system. It can be found that the hydrophobic SiO$_2$ aerogel power was easy to form aggregates from. The SiO$_2$ aerogel power has a small particle size, large specific surface area, and high surface energy, for which the particles can adhere to each other and thereby leading to aggregation in the waterborne coating system. The particle agglomeration might make the thermal insulation performance of the coating decrease. On the other hand, particle agglomeration meant that the SiO$_2$ aerogel power was incompatible with the other components in the coating formulation, which was harmful to the mechanical properties of the coating. Therefore, it is necessary to perform surface modification of the hydrophobic aerogel to maintain the structure of

"inner hydrophobic and outer hydrophilic" and to improve the dispersion uniformity in the waterborne coating system while ensuring the thermal insulation performance.

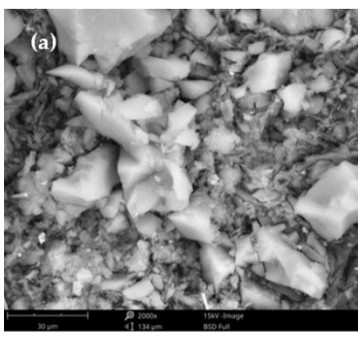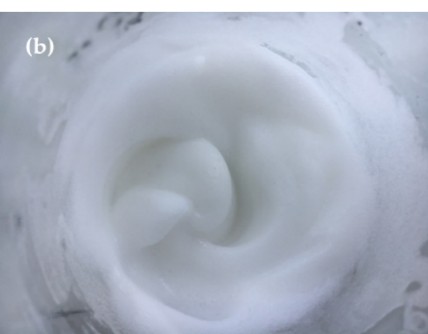

**Figure 1.** (**a**) SEM image of hydrophobic $SiO_2$ aerogel in waterborne coating system, (**b**) photo of the prepared $SiO_2$ aerogel slurry.

To improve the dispersivity of $SiO_2$ aerogel power, non-ionic surfactant and anionic dispersant were added with the unmodified $SiO_2$ aerogel power (15 wt.%), the mass fraction of the functional assistant agent was controlled at about 5%, and the rest was deionized water. The slurry had the best dispersion effect after dispersing at speed of 1000 r/min for 1 h, as shown in Figure 1b. It can be seen that the $SiO_2$ aerogel power was uniformly dispersed in the slurry, which indicated that the dispersivity of $SiO_2$ aerogel power was significantly improved after the modification.

FT-IR analysis was further carried out to verify the conversion of $SiO_2$ aerogel in the surface-modified process. As shown in Figure 2 of the unmodified $SiO_2$ aerogel, the peak at 1095 cm$^{-1}$ could be ascribed to the anti-symmetric stretching vibration of Si–O–Si, while peaks at 798 and 466 cm$^{-1}$ could be assigned to the symmetrical telescopic vibration and bending vibration of Si–O. The small absorption peaks of 1395 and 747–791 cm$^{-1}$ were the shear bending vibration and plane bending vibration of –CH$_3$, respectively, indicating that there were hydrophobic groups on the terminal branch of the aerogel [22]. After the modification with non-ionic surfactant and anionic dispersant, several peaks appeared in the spectrum compared with that of the unmodified $SiO_2$ aerogel. For example, peaks at 3450 and 1638 cm$^{-1}$ could be ascribed to the anti-symmetric stretching vibration and bending vibration of –OH, while the peak at 955 cm$^{-1}$ could be assigned to the bending vibration of Si–OH. The peak at 2900 cm$^{-1}$ could be addressed to the stretching vibration peak of C–H, which belongs to the same peak groups containing shear bending vibration peak and plane bending vibration peak of –CH$_3$ at 1395 cm$^{-1}$ and 747–791 cm$^{-1}$, respectively. The peak of 1250 cm$^{-1}$ was attributed to the C–N bond that came from 3-aminopropyl silanetriol of hydrolyzed KH550 [23]. All of these bands' intensities were enhanced, suggesting that hydrophilic groups were grafting onto the aerogel surface to improve its dispersivity in an aqueous system.

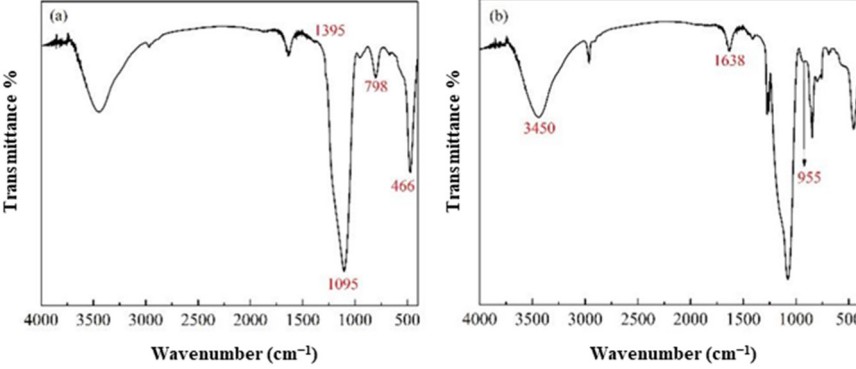

**Figure 2.** The FT-IR spectrum of $SiO_2$ aerogel power before (**a**) and after (**b**) modification.

The schematic of the modification mechanism of SiO$_2$ aerogel was as shown in Figure 3. There were both hydrophobic and hydrophilic groups in the non-ionic surfactant, and the hydrophobic groups could adsorb at the aerogel microsphere surface through van der Waals forces to reduce the surface tension of SiO$_2$ aerogel particles, which can be confirmed by the existence of a certain number of hydrophilic groups in the FT-IR spectrum of the microsphere surface. In addition, the silane coupling agent could hydrolyze to form hydroxy silane containing hydrophilic groups of Si–OH and Si–NH$_2$, which could be grafted onto the surface of the aerogel microsphere by chemical bonding to further improve the hydrophilicity. So SiO$_2$ aerogel particles could disperse evenly in a waterborne medium with high stability.

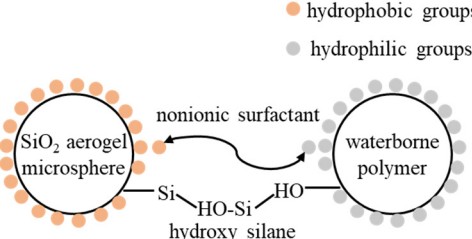

**Figure 3.** The schematic of dispersion mechanism of SiO$_2$ aerogel.

The results of contact angle measurement (in Figure S2) showed that the contact angles of the SiO$_2$ aerogel decreased from 130° to 50° after the modification, which indicated that the hydrophilic property of the hydrophobic SiO$_2$ aerogel had been enhanced. That is to say, the compatibility of the hydrophobic SiO$_2$ aerogel in the waterborne coating system was enhanced by surface modification. At the same time, such an improvement in compatibility could overcome the reunion phenomenon in the slurry greatly and lay a foundation for the preparation of SiO$_2$ aerogel thermal insulation coatings. As shown in Figure 4, there was no obvious aggregation of SiO$_2$ aerogel power was observed and the power dispersed evenly in the waterborne coating system.

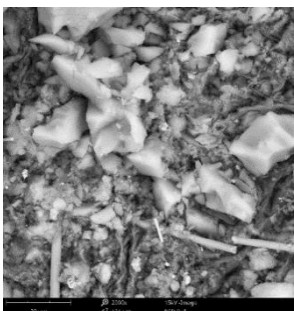

**Figure 4.** SEM image of modified SiO$_2$ aerogel in the waterborne coating system.

### 3.2. Modulation of Thermal Insulation and Mechanical Property

#### 3.2.1. The Effects of Fibers on Coating Properties

Through experiments and theoretical calculations, previous researchers found that reinforcing fibers could enhance the cracking resistance of thermal insulation coatings [24–27]. With increasing the addition of fibers, the shrinkage of thermal insulation coatings decreased while the elastic modulus increased, which could greatly improve the mechanical property of the coatings. Generally speaking, there is no particular limitation on the fibers suitable for use during the development of thermal insulation coatings, including but not limited to ceramic fibers, basalt fibers, glass fibers, mineral fibers, polyester fibers, etc. However, the improvement effects of different fibers on the mechanical property and thermal insulation properties of the coatings are different [28,29]. Therefore, adding more than one type of fiber to the thermal insulation coatings was a more recommended solution.

In this work, glass fibers (length was between 1–3 mm) and ceramic fibers (length was below 1 mm) were selected. The glass fibers have low thermal conductivity and large thermal expansion coefficient, which was a benefit to ensuring the insulating performance and improving the shrinkage and cracking resistance properties [30]. While the zirconium containing ceramic fibers can effectively reduce the crystallization process at high temperatures and delay the sintering effect at fiber interleaving contact pinto. Therefore, it is ensured that the coating still has good mechanical strength [16,31,32].

Figure 5 shows that the length of fibers had a great influence on the preparation and property of the coating. When ceramic fibers with excessively short lengths were added, obvious cracks were observed in the coating, which meant that the cracking resistance of the coating was adversely affected. While if the fibers were too long, the workability of the coatings would be adversely affected. When fibers with different lengths were added, as shown in Figure 5b, the adverse effects on the surface cracking resistance and workability were eliminated, the prepared coating showed a relatively even surface without cracking.

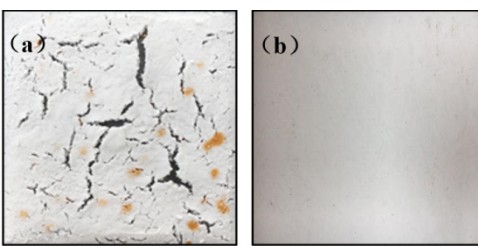

**Figure 5.** (**a**) coating morphology with only ceramic fibers (length was below 1 mm); (**b**) coating morphology with mixed fibers of different lengths (length was between 1–3 mm).

The microstructure of fiber-reinforced $SiO_2$ aerogel thermal insulation coating was shown in Figure S3. The coating surface was in a dense state, without cracks. The white bright spots were evenly distributed, indicating that the hollow glass beads were evenly distributed, which would greatly hinder heat conduction through the air and ensure the excellent heat insulation effect [33]. The inner microstructure of the coating was loose and porous. The disorderly arranged fibers connected firmly with the surrounding matrix (hollow glass beads, $SiO_2$ aerogel powers, and resin), which therefore could reduce the changes caused by film-forming shrinkage and temperature difference, while making the coating have excellent crack resistance and bond strength.

The thermal conductivity (λ) and tensile strength of the coating with or without fibers were tested, and the results were shown in Table 2. The coatings combined with fibers could encapsulate substantial void space to exhibit low thermal conductivity. In addition, the tensile strength of the coating was significantly enhanced because of the addition of fibers too.

**Table 2.** Thermal conductivity of the coating with or without fibers.

| Items | with Fibers | Without Fibers |
|---|---|---|
| $\lambda/[\text{W}\cdot(\text{m}\cdot\text{k})^{-1}]$ | 0.051 | 0.069 |
| surface topography | no cracks | full of cracks |
| tensile strength/kPa | 1520 | 450 |

### 3.2.2. The Effects of Mass Ratio of Silica Aerogel and Hollow Glass Microsphere on Coating Properties

The coatings with various mass ratios of silica aerogel and hollow glass microspheres were prepared by changing the content of silica aerogel microspheres, and the difference between surface temperature and hotplate temperature (ΔT), thermal conductivity, and adhesive strength of the coatings were measured, and the results were presented in Table 3.

**Table 3.** Effect of heat insulation filler ratio on coating properties.

| Items | $w$ (SiO$_2$ Aerogel) : $w$ (Hollow Glass Beads) | | | | |
|---|---|---|---|---|---|
| | 0:20 | 5:15 | 10:10 | 15:5 | 20:0 |
| $\Delta T/°C$ | 31 | 39 | 45 | 48 | 52 |
| $\lambda/[W \cdot (m \cdot k)^{-1}]$ | 0.082 | 0.065 | 0.050 | 0.043 | 0.036 |
| Adhesive strength/kPa | 1800 | 1530 | 1024 | 618 | 407 |

As shown in Table 3, with the mass fraction of silica aerogel increased, the thermal conductivity decreased and the difference between surface temperature and hotplate temperature increased relatively. This was because the uniformly dispersed silica aerogel microspheres in the coatings degraded the thermal conductivity by extending the length of the heat-transfer path [34–38]. Results showed that the aerogels had better thermal insulation performance than hollow glass microspheres. However, with the mass fraction of silica aerogel increased, the adhesive strength of the coating decreased. The aerogel SiO$_2$ was porous which was adverse to forming a dense coating structure in the coating. Overall, when the mass ratio of hollow glass to SiO$_2$ aerogel microspheres was 1:1, the overall performance of the coating was the best.

3.2.3. The Effects of P/B ratio on Coating Properties

The addition of thermal insulation fillers can effectively enhance thermal insulation performance, while with the number of thermal insulation fillers increasing, the mechanical performance of the coating would be reduced in turn. Solving the contradiction between thermal insulation performance and mechanical performance was a key process to obtain a perfect thermal insulation coating formula. Therefore, it was necessary to find a balance between the two in the formula design. The effect of the ratio of pigments to film former on coating property was analyzed. The test results were shown in Table 4, and the images of the samples after the adhesive strength test was provided in Figure S4.

**Table 4.** Thermal conductivity and adhesive strength of the coating with various P/B ratios.

| Test Item | P/B Ratio | | | | |
|---|---|---|---|---|---|
| | 3:1 | 2:1 | 1:1 | 1:2 | 1:3 |
| $\lambda/[W \cdot (m \cdot k)^{-1}]$ | 0.039 | 0.043 | 0.050 | 0.058 | 0.063 |
| Adhesive strength /kPa | 342 | 501 | 1024 | 1211 | 1460 |
| Tensile strength/kPa | 408 | 610 | 1580 | 2010 | 2600 |

As shown in Table 4, the thermal conductivity and adhesive strength of the coating increased with the decrease in the P/B ratio. A possible reason for this phenomenon is that, with the decreasing P/B ratio, more resins were contained in the coating relatively, which could provide a better capping effect on pigments and fillers, and the corresponding coating system had the higher compactness, namely the better mechanical performance. On the contrary, with the decreasing P/B ratio, the addition amount of thermal insulation fillers decreased, and the porosity per unit area decreased accordingly, which led to the reduction of the thermal insulation effect. When the P/B ratio was 1:1, the thermal insulation coating had a thermal conductivity of 0.050 W/(m·K) and adhesive strength of 1024 kPa, showing the best comprehensive performance.

**4. Conclusions**

In this study, the thermal insulation coatings were prepared using hydrophobic silica aerogel microspheres, and the factors of the comprehensive properties of the coatings, including the surface modification method and mechanism of SiO$_2$ aerogel, fibers, a mass ratio of silica aerogel, hollow glass microsphere, and P/B ratio, were discussed.

(1) There was no adverse effect on the surface cracking and workability with mixed fibers of different lengths added. The coating surface was in a dense state, without any cracks.

(2) With the mass fraction of silica aerogel increased, the thermal conductivity decreased and the difference between surface temperature and hotplate temperature as well as the adhesive strength increased relatively. When the mass ratio of hollow glass to $SiO_2$ aerogel microspheres was 1:1, the overall performance of the coating was the best.

(3) The adhesive strength between the coating and the substrate increased as the P/B radio increased. When the P/B ratio was 1:1, the thermal conductivity and adhesive strengths were 0.050 W/(m·K) and 1024 kPa, respectively, exhibiting balanced thermal insulation and mechanical properties.

**Supplementary Materials:** The following are available online at https://www.mdpi.com/article/10.3390/coatings12101421/s1, Figure S1: The photo of the device to measure the temperature difference, Figure S2: The contact angels of $SiO_2$ aerogel power before (left) and after (right) modification, Figure S3: Microstructure of fiber reinforced $SiO_2$ aerogel thermal insulation coating: (a) internal metallographic diagram of coating; (b) internal metallographic diagram of coating; (c) internal SEM of coating, Figure S4: The mages of the samples after the pull-off test. (a) P/B = 3:1, (b) P/B = 2:1, (c) P/B= 1:1, (d) P/B = 1:2, (e) P/B = 1:3, Table S1: Thermal insulation coating formula.

**Author Contributions:** Conceptualization, Z.D. and S.M.; Data curation, H.W. and Y.Q.; Formal analysis, Z.D. and S.M.; Funding acquisition, H.W. and Z.G.; Investigation, Z.G.; Methodology, H.W., B.L. and Y.Q.; Project administration, Z.D., H.W. and Y.J.; Resources, B.L.; Supervision, S.M. and Y.J.; Writing—original draft, H.W.; Writing—review & editing, Z.G. All authors have read and agreed to the published version of the manuscript.

**Funding:** This research was financially supported by the Key Projects of CNOOC Energy Development Co., Ltd. (HFZDGGFX-CZY-2020-03), the Changzhou Leading Innovative Talents Introduction and Cultivation Project (CQ20200045, CQ20210028) and Jiangsu Postdoctoral Research Funding Program (2021K528C).

**Institutional Review Board Statement:** Not applicable.

**Informed Consent Statement:** Not applicable.

**Data Availability Statement:** The authors confirm that the data supporting the findings of this study are available within the article.

**Conflicts of Interest:** The authors declare no conflict of interest.

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
