# Peer review of "Modulation of Thermal Insulation and Mechanical Property of Silica Aerogel Thermal Insulation Coatings"

_coatings, doi:10.3390/coatings12101421_

Round 1
Reviewer 1 Report
Modulation of Thermal Insulation and Mechanical Property of Silica Aerogel Thermal Insulation Coatings
In this work, efficient silicon dioxide aerogel thermal insulation coatings were obtained via the high thermal isolation ability of SiO2 aerogel. Scanning electron microscope and FT-IR techniques as well as contact angle and temperature difference tests were used to examine the properties of thermal isolation mechanism. Despite efforts by authors, the reviewer has some comments on the work.
Away from only four references (3, 10, 19, and 21) out of 31 reference, the used references are not up to date.
In the abstract, it has not mentioned that the silica was hydrophobic as happened in the conclusion.
It is useful to put some numbers on points 1, and 2 of the conclusions.
Write about the contact angle measurement device.
Contact angle measurements is a contamination sensitive process. Write about your surface cleaning strategies to avoid any dramatic error of measurements.
Figure 2 contains very Astonishing images but unfortunately the details in these images not adequately described.
Table of results of contact angle measurements is missing.
What was the standard deviation of your measurements and how many time each test was repeated?
English language polishing is intensively required.
Away from that, a nice piece of work.
Author Response
Reviewer 1
In this work, efficient silicon dioxide aerogel thermal insulation coatings were obtained via the high thermal isolation ability of SiO2 aerogel. Scanning electron microscope and FT-IR techniques as well as contact angle and temperature difference tests were used to examine the properties of thermal isolation mechanism. Despite efforts by authors, the reviewer has some comments on the work.
- Away from only four references (3, 10, 19, and 21) out of 31 reference, the used references are not up to date.
Response: More up-to-date references were added in the revised manuscript.
- In the abstract, it has not mentioned that the silica was hydrophobic as happened in the conclusion.
Response: The raw material of silica was hydrophobic, while after modification, the silica was hydrophilic.
- It is useful to put some numbers on points 1, and 2 of the conclusions.
Response: Numbers on points were put in the conclusions
- Write about the contact angle measurement device.
Response: The contact angel was measured by contact angel tester (OCA25, Dataphysics Instruments GmbH, Germany), which was mentioned in 2.3 Performance tests.
- Contact angle measurements is a contamination sensitive process. Write about your surface cleaning strategies to avoid any dramatic error of measurements.
Response: During the measurement, we controlled the droplet size and velocity to avoid dramatic error of measurements
- Figure 2 contains very Astonishing images but unfortunately the details in these images not adequately described.
Response: The description and discussion of Figure 2 (SEM) were improved in the revised manuscript.
- Table of results of contact angle measurements is missing.
Response: In the contact angle measurements section, there were only two contact angle values of SiO2 aerogel powers before (130°) and after (50°) modification were measured, and the results were described in the paragraph. So we thought it was not necessary to list the results in a table.
- What was the standard deviation of your measurements and how many time each test was repeated?
Response: Standard deviation of the results were added. And all the test were at least repeated three times.
- English language polishing is intensively required.
Response: Thanks for the suggestion, we had made some modifications in English language.
Away from that, a nice piece of work.

Reviewer 2 Report
The manuscript "coatings-1821438" by Di et al. reported Modulation of Thermal Insulation and Mechanical Property of Silica Aerogel Thermal Insulation Coatings. After review, this study is interesting but currently unpublishable. The authors have to make major changes. The authors should refer to the following comments to improve their work:
General comments:
1. Please improve the quality of all Figures.
2. Please add or replace more up-to-date references.
3. Please improve the discussion.
4. Move Table 2 and Figures 1, 5, and 8 to the Supporting Information.
Specific comments:
1. In fact, the abstract is something which is showing the work to reader, so please improve the abstract and provide more complete and accurate results.
2. Terms such as SiO2 (line 9), SEM (line 11), and FT-IR (line 11) are abbreviated in the abstract without explanation, please revise and rewrite.
3. Please use the abbreviation of Silica aerogel (SiO2 aerogel) in the introduction (Lines 44,47, and 57).
4. Please review the results of several other studies in the introduction and explain the difference between this study and the others.
5. Report the materials used in this work in one paragraph and please remove Table 1.
6. Please explain more clearly about the preparation of SiO2 airgel slurry. Mention the values used.
7. The titles of section 2.2.1. and part 3.1. are the same. There are many errors. The manuscript seems to have been written hastily. Please do a general review. The language of the manuscript should also be checked.
8. Please distinguish the images in Figure 2, also put Figure 2 after line 127.
9. Please reconsider the description of Figure 2 (lines 120 to 127). Please cite the reference. In the sentence "It can be found from Figure 1 that hydrophobic SiO2 aerogel power aggregates to gether" (line 121) do you mean Figure 2?
10. Please specify in Figure 4 which spectrum is before and after the change.
11. Place Figure 4 after line 149.
12. In Figure 4b, what is the reason for the increase in the intensity of the peak around 2900 and which group is it attributed? Also, the peak that appeared around 1250 is attributed to which group and what is the reason for its appearance?
13. Please specify in Figure 6 which image is before and after the change.
14. What is the reason for using Figure 7? In fact, Figure 7 is one of the SEM images shown in Figure 2 (it is clear that the image in Figure 2 is cropped).
15. Please provide images of the sample prepared for the pull-off test (in the Supplementary Information).
Author Response
Reviewer 2
The manuscript "coatings-1821438" by Di et al. reported Modulation of Thermal Insulation and Mechanical Property of Silica Aerogel Thermal Insulation Coatings. After review, this study is interesting but currently unpublishable. The authors have to make major changes. The authors should refer to the following comments to improve their work:
General comments:
- Please improve the quality of all Figures.
Response: We had replace some Figures with low quality in the revised manuscript.
- Please add or replace more up-to-date references.
Response: More up-to-date references were added in the revised manuscript.
- Please improve the discussion.
Response: The discussion section was improved in the revised manuscript.
- Move Table 2 and Figures 1, 5, and 8 to the Supporting Information.
Response: All the tables and figures were important to describe the research process and support the results. And on the other hand, the length of this paper was not that long. So we thought it was necessary to attach a Supporting Information file.
Specific comments:
- In fact, the abstract is something which is showing the work to reader, so please improve the abstract and provide more complete and accurate results.
Response: The abstract was improved in the revised manuscript.
- Terms such as SiO2(line 9), SEM (line 11), and FT-IR (line 11) are abbreviated in the abstract without explanation, please revise and rewrite.
Response: The full name of the abbreviations were provided.
- Please use the abbreviation of Silica aerogel (SiO2aerogel) in the introduction (Lines 44,47, and 57).
Response: The “Silica aerogel” was corrected to “SiO2 aerogel” in the revised manuscript.
- Please review the results of several other studies in the introduction and explain the difference between this study and the others.
Response: Relevant information had been added in the introduction section in the revised manuscript.
- Report the materials used in this work in one paragraph and please remove Table 1.
Response: In this work, more than 10 types of material were used, and we thought that it could be easier for readers to figure out and obtain the information in table than that in text description.
- Please explain more clearly about the preparation of SiO2 aerogel slurry. Mention the values used.
Response: More detail information about the preparation of SiO2 aerogel slurry was added in the revised manuscript.
- The titles of section 2.2.1. and part 3.1. are the same. There are many errors. The manuscript seems to have been written hastily. Please do a general review. The language of the manuscript should also be checked.
Response: The title of 3.1 was modified. And we have tried our best to improved the language of the manuscript.
- Please distinguish the images in Figure 2, also put Figure 2 after line 127.
Response: The two SEM photos were marked as (a) and (b) in Figure 3 (previous Figure 2), and the Figure was after line 127.
- Please reconsider the description of Figure 2 (lines 120 to 127). Please cite the reference. In the sentence "It can be found from Figure 1 that hydrophobic SiO2 aerogel power aggregates to gether" (line 121) do you mean Figure 2?
Response: Yes, it meant Figure 2 (namely Figure 2a in the revised manuscript)
- Please specify in Figure 4 which spectrum is before and after the change.
Response: Spectra before and after the change were be specified in the revised manuscript
- Place Figure 4 after line 149.
Response: Figure 4 was put after line 149.
- In Figure 4b, what is the reason for the increase in the intensity of the peak around 2900 and which group is it attributed? Also, the peak that appeared around 1250 is attributed to which group and what is the reason for its appearance?
Response: The peak around 2900 is attributed to stretching vibration peak of C-H, peak that appeared around 1250 is attributed to C-N bond. All the peaks increased meant that hydrophilic groups grafted onto SiO2 aerogel surface
- Please specify in Figure 6 which image is before and after the change.
Response: Spectra before and after the change were be specified in the revised manuscript
- What is the reason for using Figure 7? In fact, Figure 7 is one of the SEM images shown in Figure 2 (it is clear that the image in Figure 2 is cropped).
Response: Figure 2 is the SEM image of hydrophobic SiO2 aerogel, while Figure 7 (Figure 6 in the revised manuscript) is the SEM image of modified SiO2 aerogel.
- Please provide images of the sample prepared for the pull-off test (in the Supplementary Information).
Response: The images of the sample that before and after pull-off test as follows. We thought the quality of these images were too low to present in the article, so we just provide the test results in Tables.

Reviewer 3 Report
The application of silica aerogels for thermal insulating od one of the most important application of these materials. This is the reason why there is a lot of works devoted to this subject. One of them is the reviewed paper. Unfortunatelly, the amount of novelty in the manuscript is very low. Authors observed that the addition of fibers improves mechanical properties of insulation (one maight expect that) and that the thermal conductivity decrease with the increase of aerogel fraction (this is also not a surprise while the silica aerogel has one of the smallest value of thermal conductivity coefficient). The whole conclusion section is 14 lines long and does not contain aby informations which are not obvious for anyone who has at least basic knowledge about the aerogels and insulation. Some minor remarks: 1. Introduction should be rewritten. In current version there is a lot of banalities like e.g. "insulation coatings are widely used for passive thermal insulation" (page 1, line 29); and - in very next sentence! - "The coating works by insulating". The information that insulation insulates - is not this kind of konwledge that the Reader needs to understand of motivation of the paper. 2. The language definitelly needs proofreading and mamy corrections. I am not a native speaker but even to me it is hard to accept such sentences as e.g. "The materials were used in this work were listed in Table 1." There is also a lot of typos: my favorite: "contact angel " (page 3, line 87) and "P/B radio" (page 9) 3. The description of the materials is highly insufficient. Authors list, among others, "wetting agent", "defoaming agent" and some other "agents" - but they do not precise what kind of substances they are. Similarly, when introducing ceramic fibers, we do not obtain informations about e.g. the mean diameter of these fibers. Finally, the deionized water is described simply as "made by the laboratory" (without informations about conductivity or ionic strength). This makes the results of authors impossible to repeat or verify by another team. 4. Figures are flawed: - Fig. 1: contrary to the caption, this is not "schematic diagram of temperature difference" but the photo of device measuring (probabely) the temperature... One does not know where because it is not shown in this photo - Fig. 3 does not present any informations - Fig. 4 - the vertical axis does not have labels The application of silica aerogels for thermal insulating od one of the most important application of these materials. This is the reason why there is a lot of works devoted to this subject. One of them is the reviewed paper. Unfortunatelly, the amount of novelty in the manuscript is very low. Authors observed that the addition of fibers improves mechanical properties of insulation (one maight expect that) and that the thermal conductivity decrease with the increase of aerogel fraction (this is also not a surprise while the silica aerogel has one of the smallest value of thermal conductivity coefficient). The whole conclusion section is 14 lines long and does not contain aby informations which are not obvious for anyone who has at least basic knowledge about the aerogels and insulation. Some minor remarks: 1. Introduction should be rewritten. In current version there is a lot of banalities like e.g. "insulation coatings are widely used for passive thermal insulation" (page 1, line 29); and - in very next sentence! - "The coating works by insulating". The information that insulation insulates - is not this kind of konwledge that the Reader needs to understand of motivation of the paper. 2. The language definitelly needs proofreading and mamy corrections. I am not a native speaker but even to me it is hard to accept such sentences as e.g. "The materials were used in this work were listed in Table 1." There is also a lot of typos: my favorite: "contact angel " (page 3, line 87) and "P/B radio" (page 9) 3. The description of the materials is highly insufficient. Authors list, among others, "wetting agent", "defoaming agent" and some other "agents" - but they do not precise what kind of substances they are. Similarly, when introducing ceramic fibers, we do not obtain informations about e.g. the mean diameter of these fibers. Finally, the deionized water is described simply as "made by the laboratory" (without informations about conductivity or ionic strength). This makes the results of authors impossible to repeat or verify by another team. 4. Figures are flawed: - Fig. 1: contrary to the caption, this is not "schematic diagram of temperature difference" but the photo of device measuring (probabely) the temperature... One does not know where because it is not shown in this photo - Fig. 3 does not present any informations - Fig. 4 - the vertical axis does not have labels.
Author Response
Reviewer 3
The application of silica aerogels for thermal insulating is one of the most important application of these materials. This is the reason why there is a lot of works devoted to this subject. One of them is the reviewed paper. Unfortunately, the amount of novelty in the manuscript is very low. Authors observed that the addition of fibers improves mechanical properties of insulation (one might expect that) and that the thermal conductivity decrease with the increase of aerogel fraction (this is also not a surprise while the silica aerogel has one of the smallest value of thermal conductivity coefficient). The whole conclusion section is 14 lines long and does not contain any information which are not obvious for anyone who has at least basic knowledge about the aerogels and insulation. Some minor remarks:
Introduction should be rewritten. In current version there is a lot of banalities like e.g. "insulation coatings are widely used for passive thermal insulation" (page 1, line 29); and - in very next sentence! - "The coating works by insulating". The information that insulation insulates - is not this kind of knowledge that the Reader needs to understand of motivation of the paper.
Response: Introduction has be rewritten in the revised manuscript.
- The language definitelyneeds proofreading and many corrections. I am not a native speaker but even to me it is hard to accept such sentences as e.g. "The materials were used in this work were listed in Table 1." There is also a lot of typos: my favorite: "contact angel " (page 3, line 87) and "P/B radio" (page 9)
Response: We are so sorry for the poor English. In the revised manuscript, we have made obvious changes in language.
- The description of the materials is highly insufficient. Authors list, among others, "wetting agent", "deforming agent" and some other "agents" - but they do not precise what kind of substances they are. Similarly, when introducing ceramic fibers, we do not obtain information about e.g. the mean diameter of these fibers. Finally, the deionized water is described simply as "made by the laboratory" (without information about conductivity or ionic strength). This makes the results of authors impossible to repeat or verify by another team.
Response: Detail information of the materials used in this work were provided in Table 1.
- Figures are flawed:
- 1: contrary to the caption, this is not "schematic diagram of temperature difference" but the photo of device measuring (probabely) the temperature... One does not know where because it is not shown in this photo
Response: Relative changes have been made in the revised manuscript.
- 3 does not present any information
Response: The Figure 3 present the macroscopic photo of the prepared SiO2 aerogel slurry, and it can be seen that the SiO2 aerogel was uniformly dispersed in the slurry. The related information was added in the revised manuscript. At the same time, the Figure 3 was put before the Figure 2.
- 4 the vertical axis does not have labels
Response: Vertical axis of FT-IR were relative values.

Round 2
Reviewer 2 Report
1. Please improve the discussion and don't just report data.
2. Move Table 2 and Figures 1, 5, and 8 to the Supporting Information.
3. Please provide images of the sample prepared for the pull--off test (in the Supplementary Information).
4. The language of the manuscript should be checked.
Author Response
- Please improve the discussion and don't just report data.
Response: We have made some improvements in the discussion section in the revised manuscript.
- Move Table 2 and Figures 1, 5, and 8 to the Supporting Information.
Response: Table 2 and Figures 1, 5, and 8 has been moved to the Supporting Information in the revised manuscript.
- Please provide images of the sample prepared for the pull--off test (in the Supplementary Information).
Response: The images of the samples after pull-test were provided in the Supplementary Information as Figure S4.
- The language of the manuscript should be checked.
Response: In the revised manuscript, we have checked and improved the language.

Reviewer 3 Report
The manuscript has been significantly improved. The Authors have rewritten the introduction, added more informations about materials and methods and posted some actual conclusions from their research (mainly information about the optimal mass ratio of hollow glass to SiO2 aerogel microspheres). Thus, though the still low amount of novelty of the presented material, I reccomend the manuscript for publication. It may be of some interest as one of the many contributions to improving the properties of aerogel insulations.
Author Response
Thanks for your kindly comments and suggestions.